# MIMONets: Multiple-Input-Multiple-Output Neural Networks Exploiting Computation in Superposition

**Nicolas Menet**[1,2*]
menetn@ethz.ch

**Michael Hersche**[1,2]
her@zurich.ibm.com

**Geethan Karunaratne**[1]
kar@zurich.ibm.com

**Luca Benini**[2]
lbenini@iis.ee.ethz.com

**Abu Sebastian**[1]
ase@zurich.ibm.com

**Abbas Rahimi**[1]
abr@zurich.ibm.com

[1]IBM Research – Zurich, [2]ETH Zurich

## Abstract

With the advent of deep learning, progressively larger neural networks have been designed to solve complex tasks. We take advantage of these capacity-rich models to lower the cost of inference by exploiting *computation in superposition*. To reduce the computational burden per input, we propose Multiple-Input-Multiple-Output Neural Networks (MIMONets) capable of handling many inputs at once. MIMONets augment various deep neural network architectures with variable binding mechanisms to represent an arbitrary number of inputs in a compositional data structure via fixed-width distributed representations. Accordingly, MIMONets adapt nonlinear neural transformations to process the data structure holistically, leading to a speedup nearly proportional to the number of superposed input items in the data structure. After processing in superposition, an unbinding mechanism recovers each transformed input of interest. MIMONets also provide a dynamic trade-off between accuracy and throughput by an instantaneous on-demand switching between a set of accuracy-throughput operating points, yet within a single set of fixed parameters. We apply the concept of MIMONets to both CNN and Transformer architectures resulting in MIMOConv and MIMOFormer, respectively. Empirical evaluations show that MIMOConv achieves $\approx 2\text{--}4\times$ speedup at an accuracy delta within $[+0.68, -3.18]\%$ compared to WideResNet CNNs on CIFAR10 and CIFAR100. Similarly, MIMOFormer can handle 2–4 inputs at once while maintaining a high average accuracy within a $[-1.07, -3.43]\%$ delta on the long range arena benchmark. Finally, we provide mathematical bounds on the interference between superposition channels in MIMOFormer. Our code is available at `https://github.com/IBM/multiple-input-multiple-output-nets`.

## 1  Introduction

Driven by the successes of deep learning in image and natural language processing tasks, increasingly large neural network models have been developed to reach state-of-the-art performance [1–4]. These large models, however, increase computational complexity in terms of operation count for every event of input processing. One viable option to reduce the computational cost of processing per input is to create a compositional data structure where a variable number of input items (i.e., values) can be bound to corresponding protection keys, creating key-value pairs that can coexist and be processed

---

*Research conducted at IBM Research – Zurich.

37th Conference on Neural Information Processing Systems (NeurIPS 2023).

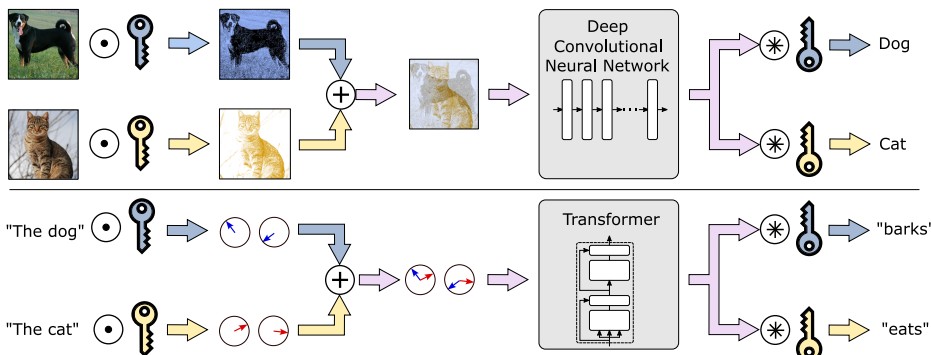

Figure 1: MIMONets simultaneously pass multiple inputs through a nonlinear function, e.g., a deep convolutional network (on top) or a Transformer (on bottom). Input samples are bound with high-dimensional keys to project the samples into quasi-orthogonal subspaces. The results of the individual samples are retrieved at the end of the network by unbinding with corresponding keys.

concurrently. This variable-sized data structure can be represented by fixed-width distributed representations in vector-symbolic architectures (VSAs) [5–7]. In VSAs, the composition of different items in the data structure is based on functional compositionality (i.e., key-value binding), which yields a dimensionality-preserving distributed representation, rather than concatenative compositionality. Interestingly, the resulting fixed-width distributed data structure can be transformed by a one-time application of a function, whereby all input items are jointly transformed, leading to *holistic transformation* or *computation in superposition* [8–10]. This concept of computation in superposition can reduce the effective number of operations per input by a factor of the number of input items in the data structure, because the function is applied to the data structure holistically without decomposing the constituent items for individual transformations. However, processing the VSA data structure via computation in superposition has so far been limited to linear maps [8–10].

Motivated by these observations, we make the following contributions:

(1) We introduce a principled and transparent approach to Multiple-Input-Multiple-Output Neural Networks (MIMONets) based on VSA, enabling computation in superposition for highly nonlinear transformations in neural networks (Section 2). The MIMONets concept can be applied to various architectures, embracing the rich capacity provided by increasingly large models. The resulting network can handle many inputs at once, thus reducing the computational cost per input. We describe and overcome the challenges of computation in superposition, which originates from nonlinear interference of inputs in separate superposition channels.

(2) We propose MIMOConv, a realization of MIMONets for deep convolutional neural network (CNN) architectures (Section 3). We provide several strategies for mitigating interference between different superposition channels, including a novel locality-preserving binding operation (PWHRR) and isometry-inducing regularization. Empirical evaluations on CIFAR10 and CIFAR100 show that a MIMOConv built on a WideResNet-28-10 [11] can process concurrently two inputs in superposition ($\approx 2\times$ speedup) even with a slightly higher accuracy (0.11–0.68% gain), and four inputs ($\approx 4\times$ speedup) with a marginal drop (1.24–3.18%) (see Section 5.1).

(3) We further extend the concept of MIMONets to Transformer architectures, where the calculation of attention scores poses additional challenges for the computation in superposition paradigm. To this end, we propose MIMOFormer, which relies on a 2D grid binding scheme for computing attention in superposition (Section 4). We derive probabilistic tail bounds on the distortion caused by interchannel interference and show that our method converges to noise-free attention in the limit of high dimension. Our method succeeds empirically ($\geq 96.52\%$ accuracy) at synthetic sequence modelling tasks [12], while previous work [13] fails ($\leq 20.04\%$). We also provide evaluations on the long range arena (LRA) dataset [14] in Section 5.2 using a MIMOFormer that is based on the Performer architecture [15]. Compared to the Performer, MIMOFormer maintains a high average accuracy with a marginal drop of 1.07% and 3.43% when handling two and four inputs at once, respectively.

(4) MIMONets allow a *dynamic* trade-off at inference time between accuracy and speed, i.e., they offer an instantaneous on-demand switching between accuracy-throughput operating points using *a*

*single set of fixed parameters*. Experimental results in Section 5.1 show that a dynamic MIMOConv can seamlessly operate at various modes ($\approx 1-4\times$ speedup) while maintaining a high accuracy compared to the best static models ($\leq 1.82\%$ drop).

## 2 MIMONets enabling Computation in Superposition

The central idea behind MIMONets (see Figure 1) is to simultaneously pass multiple inputs as a superposition through a nonlinear function $f_\theta$ parameterized by neural network weights $\theta$. We isolate the individual inputs into separate protected *channels* by binding them with protection keys resulting in a key-value data structure [5–7].

**Concept.** Assuming two inputs ($x^{(1)}$ and $x^{(2)}$), which can be generic embeddings either from images or natural language, we define a unique high-dimensional key ($a^{(1)}$ and $a^{(2)}$) for each protected channel, drawn randomly at initialization. Owing to the Blessing of Dimensionality, randomly drawn high-dimensional vectors are quasi-orthogonal with high probability (see Appendix A). Consequently, binding ($\odot$) the inputs with these keys yields quasi-orthogonal key-value pairs ($x^{(1)} \odot a^{(1)}$ and $x^{(2)} \odot a^{(2)}$), which enables one to superpose the pairs with low interference:

$$s = a^{(1)} \odot x^{(1)} + a^{(2)} \odot x^{(2)}. \tag{1}$$

As discussed in Appendix A, $s$ admits a noisy retrieval of $x^{(1)}$ and $x^{(2)}$ through unbinding:

$$\hat{x}^{(1)} = a^{(1)} \circledast s = a^{(1)} \circledast a^{(1)} \odot x^{(1)} + a^{(1)} \circledast a^{(2)} \odot x^{(2)} = x^{(1)} + noise \tag{2}$$

To accelerate computing, inspired by the above-mentioned noisy retrieval, we pass the superposition $s$ through a nonlinear function $f_\theta$ with parameters $\theta$, such as a neural network, before retrieval. The quasi-orthogonality of the bound inputs allows processing each in a separate *protected* subspace—all with a single function call. To be able to recover the first processed sample $f_\theta(x^{(1)})$ from $f_\theta(s)$, we aim to find an unbinding key $\tilde{a}^{(1)}$ for which

$$\tilde{a}^{(1)} \circledast f_\theta(s) \approx \tilde{a}^{(1)} \circledast f_\theta \left( a^{(1)} \odot x^{(1)} \right) + \tilde{a}^{(1)} \circledast f_\theta \left( a^{(2)} \odot x^{(2)} \right) \tag{3}$$

$$\approx f_\theta \left( x^{(1)} \right) + \tilde{a}^{(1)} \circledast f_\theta \left( a^{(2)} \odot x^{(2)} \right). \tag{4}$$

The first approximation holds exact for linear $f_\theta$. As discussed in Section 3, a nonlinear $f_\theta$ can still be encouraged to allow such an approximation through appropriate weight regularization techniques and well-suited activation functions. Further, by optimizing over unbinding keys ($\tilde{a}^{(i)}$), the second estimation (Eq. (4)) can be achieved. Consequently, matching binding and unbinding keys ($a^{(i)}$ and $\tilde{a}^{(i)}$) that confirm the approximation (Eq. (3) and (4)) set up a protected *channel* through the nonlinear function $f_\theta(s)$. Appendix A lists the design choices of the adopted VSA, which define the operations of key-value binding and unbinding, for all MIMONet variants presented in this work. In the case of image embeddings, we use circular convolution [6] for binding and Matrix Binding of Additive Terms (MBAT) [16] for unbinding. In the case of sequence tokens, we bind and unbind using the Hadamard product [17]. Binding and unbinding keys are always data-independent, i.e., they depend only on the index of the protected channel. See [18] for alternative binding and unbinding options.

**Dynamic Inference.** Setting up $N$ protected channels through a neural network $f_\theta$ gives almost a speedup of $N\times$ due to most computations taking place in superposition. However, as is explored empirically, increasing $N$ adds inter-channel noise leading to a decrease in predictive accuracy. If a fixed trade-off is unsatisfactory, one can build a dynamic model capable of running a superposition of one up to $N$ different inputs. By inserting the same input into multiple channels and averaging the output, one effectively forms an in-network ensemble, similar to [19, 20]. Using all protected channels for different inputs leads to a fast but less accurate model, whereas using all protected channels for the same input yields a slower but accurate ensemble model. By partitioning the superposition channels on demand, arbitrary configurations in between may be reached. Note that our method can instantaneously adapt to the current computational demand, without loading different model weights from the main memory. To perform across slow and fast configurations, the model should randomly switch between them during training. See Appendix A for a more detailed explanation.

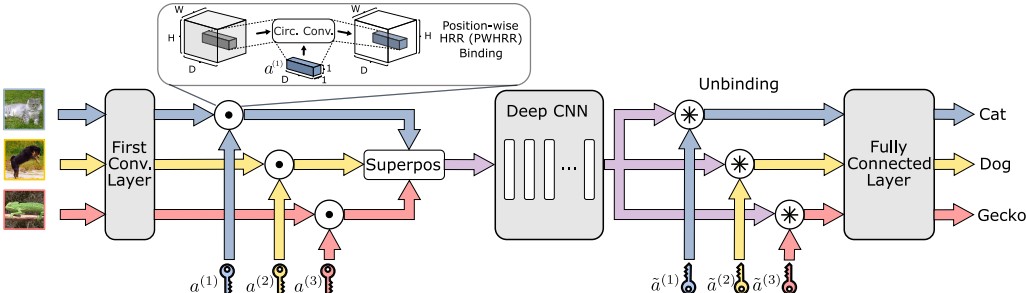

Figure 2: MIMOConv configured with $N$=3 channels. The input images are passed individually through the first convolutional layer before binding each feature value with a channel-specific high-dimensional key. The key-value pairs are superposed yielding a dimensionality-preserving composition and passed through the rest of the CNN layers. The output is unbound with corresponding keys, and the unbound representations are classified separately with a shared fully-connected layer.

## 3  MIMOConv

This section presents how the MIMONets concept, introduced in Section 2, can be applied to construct a multiple-input-multiple-output CNN (MIMOConv) capable of simultaneously processing multiple images in superposition. The MIMOConv architecture is shown in Figure 2. Multiple input samples ($N$) are passed through the first convolutional layer, bound with a unique high-dimensional key based on Holographic Reduced Representations (HRR) [6], and superposed by element-wise addition. After passing the superposed tensors through the network's main CNN layers, we obtain a combined feature vector with information on all inputs. By unbinding with separately learned keys based on MBAT [16], which amounts to a matrix multiplication, we extract the individual processed information, which is then passed through a final fully-connected layer to produce logits for classification. In the following, we introduce our three main contributions that lead to a highly accurate MIMOConv.

**Augmenting CNNs with locality-preserving variable bindings.**  We embrace a principled and transparent binding mechanism from HRR. Accordingly, binding is performed using circular convolution with a binding key of dimension $D$, drawn from an i.i.d. Gaussian distribution with zero mean and $1/D$ variance. Instead of convolving the flattened image tensor, we repeatedly apply circular convolution between the binding key and each pixel volume spanning across the feature maps ($D \times 1 \times 1$). This binding operation, which we call *position-wise HRR* (PWHRR), is translation equivariant and maintains locality, an essential property for subsequent layers with limited receptive fields. More concretely, binding and unbinding are performed as

$$(a^{(k)} \odot x^{(k)})_{:,w,h} = a^{(k)} * x^{(k)}_{:,w,h} \tag{5}$$

$$(\tilde{a}^{(k)} \circledast h)_{:,w,h} = \tilde{a}^{(k)} \cdot h_{:,w,h}, \tag{6}$$

with image tensors $x^{(k)} \in \mathbb{R}^{D \times W \times H}$, hidden representation $h \in \mathbb{R}^{D' \times W \times H}$, binding key $a^{(k)} \in \mathbb{R}^D$ and unbinding key $\tilde{a}^{(k)} \in \mathbb{R}^{D' \times D'}$. Here, $D, D', W$, and $H$ denote the hidden dimension at binding, the hidden dimension at unbinding (generally differs from $D$), image width, and image height, respectively. $*$ is the circular convolution, $\cdot$ the matrix multiplication, and $k$ indexes the superposition channel. Unbinding is applied after the global (average) pooling to reduce computational costs. The binding keys can be either learned or fixed during training (see ablation study in Appendix E).

**Embracing high dimensional embeddings.**  According to the Blessing of Dimensionality (see Appendix A), random vectors quickly become quasi-orthogonal as their dimension increases. To reduce interference between protected channels, we increase the number of feature maps by adopting Wide Residual Networks [11], the most commonly used CNN architecture to achieve state-of-the-art accuracy on CIFAR100 [21]. The input tensors are passed individually through the first convolutional layer before being superposed in a suitably high dimensional space. We set the number of feature maps after the first convolutional layer to $D$=64. This is $4\times$ more than the standard Wide-ResNet-28 [11], which results in improved training stability at a marginally higher compute cost (see Section 5.1).

**Encouraging isometric layers.** We aim to preserve the quasi-orthogonality of our protected channels as the superposition is passed through many network layers. To that end, residual connections are used and each subfunction $g_\theta$ of type (strided) spatial convolution or activation function is made approximately inner-product preserving, i.e.,

$$\langle g_\theta(x), g_\theta(y) \rangle \approx \langle x, y \rangle. \tag{7}$$

Inspired by [22], we deploy a regularization to the CNN weights and use a parametric ReLU [23] activation function, a learnable affine combination between identity and ReLU. Those adjustments lead to a near-isometric network. The regularization for the individual CNN layers is determined by

$$L(W) = \frac{\gamma}{2} \big\| Conv(W, W) - \delta^{C_o} \big\|_F^2 \qquad \text{where} \qquad \delta^{C_o}_{:,:,j,l} = I_{C_o \times C_o} \cdot \mathbb{1}_{j,l=\lfloor \frac{k}{2} \rfloor}, \tag{8}$$

$$L(W^T) = \frac{\gamma}{2} \big\| Conv(W^T, W^T) - \delta^{C_i} \big\|_F^2 \qquad \text{where} \qquad \delta^{C_i}_{:,:,j,l} = I_{C_i \times C_i} \cdot \mathbb{1}_{j,l=\lfloor \frac{k}{2} \rfloor}, \tag{9}$$

with $\gamma$ as hyperparameter. We use $L(W)$ if $C_i > C_o$, else $L(W^T)$. See Appendix B for more details.

# 4 MIMOFormer

This section presents MIMOFormer, which applies the principles of computation in superposition to dot-product self-attention [24]. Figure 3 shows a MIMOFormer layer with four protected channels, consisting of a single-head[2] attention block, a concatenation, a linear layer, an MLP, and a skip connection.

Merely superposing protected attention keys[3] and queries does not yield the desired result. As discussed in Appendix F, with scalar attention scores between pairs of tokens, vanilla dot-product attention irreversibly combines attention scores of separate protected channels, effectively blurring the attention weights. By building instead on linear Transformers [15] [25], attention scores are not collapsed to scalars, thus enabling computation in superposition.

Despite being compatible with other linear transformers (such as DPFP [25]), for concreteness we discuss changes to the Performer's FAVOR+ attention block [15]. Enabling computation in superposition, we label the block as FAVOR+S.

Given attention keys $(k_j)_{j=1}^L$, queries $(q_i)_{i=1}^L$, and values $(v_j)_{j=1}^L$, FAVOR+ estimates dot-product attention at sequence index $i$ through

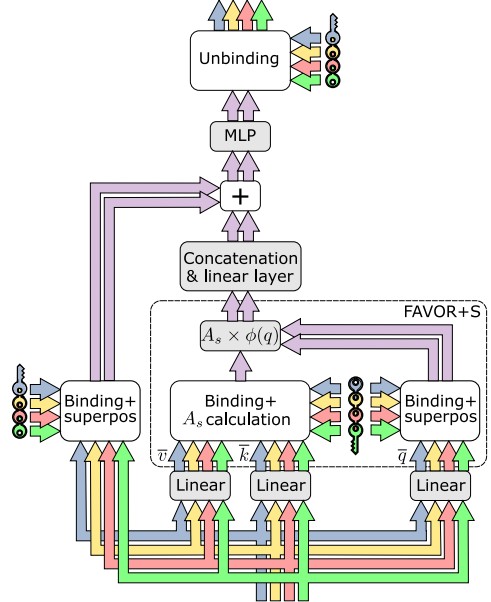

Figure 3: MIMOFormer layer applying computation in superposition to single-head FAVOR+S attention and to the MLP. This example passes four channels ($N \cdot M = 2 \cdot 2 = 4$) to the FAVOR+S attention (see Eq. (19)). The individual outputs are retrieved by unbinding after the MLP. The skip connection superposes the individual inputs for alignment, using the same protection keys as in unbinding.

$$o_i = \sum_{j=1}^L v_j \frac{\exp\big(\langle k_j, q_i \rangle / \sqrt{D}\big)}{\sum_{l=1}^L \exp\big(\langle k_l, q_i \rangle / \sqrt{D}\big)} \approx \frac{1}{B_i} \underbrace{\left[ \sum_{j=1}^L v_j \phi(k_j)^T \right]}_{A} \times \phi(q_i), \tag{10}$$

where $\phi : \mathbb{R}^D \to \mathbb{R}_+^R$ approximates the softmax kernel $\exp(\langle k_j, q_i \rangle / \sqrt{D})$ as an explicit inner product. Since computing $\phi$ has a computational complexity of $\mathcal{O}(DR)$, the construction of $A \in \mathbb{R}^{D \times R}$ takes $\mathcal{O}(LDR)$. Equally, multiplying $A \times \phi(q_i) \ \forall i$ takes $\mathcal{O}(LDR)$. Thus, FAVOR+ breaks the quadratic dependence on sequence length $L$ of attention. The denominator $B_i$ is discussed in Appendix C.

---

[2]The application to multi-head attention is straightforward; empirical results are shown in Section 5.2.

[3]Protection keys are denoted by the letter $a$ and attention keys by the letter $k$ to distinguish between them.

Our extension, FAVOR+S, computes attention in superposition, yielding a square-root speedup in the number of protected channels. It encodes them in an $M \times N$ grid, distributing the computational burden among the setup of the value-key matrix $A$ and its product with $\phi(q_i)$. Importantly, in the limit of high dimensional projections $D$ our mechanism converges to exact self-attention, completely separating the protected channels from one another. In the following, we assume the token values $\overline{v}_j$, keys $\overline{k}_j$, and queries $\overline{q}_j$ to be in protected subspaces:

$$v_j^{(m,n)} := \overline{v}_j^{(m,n)} \odot a^{(m,n)}, \qquad k_j^{(m,n)} := \overline{k}_j^{(m,n)} \odot a^{(m,n)}, \qquad q_j^{(m,n)} := \overline{q}_j^{(m,n)} \odot a^{(m,n)}, \quad (11)$$

where $a^{(m,n)}$ are i.i.d. bipolar vectors of Rademachers [17] and $(m,n)$ denotes a channel.

MIMOFormer benefits from the low time complexity ($\mathcal{O}(D)$) of the Hadamard product, especially since binding and unbinding are performed in every MIMOFormer layer. Our derivations rely on two estimates:

$$\phi(k)^T \phi(q) \overset{P}{\approx} \exp\left(\langle k, q \rangle / \sqrt{D}\right) \qquad \langle \sum_{w=1}^{N} k_j^{(u,w)}, \sum_{t=1}^{M} q_i^{(t,n)} \rangle \overset{H}{\approx} \underbrace{\langle k_j^{(u,n)}, q_i^{(u,n)} \rangle}_{\text{intended signal}} \qquad (12)$$

The approximation $P$, which improves with increasing $R = \dim(\phi(\overline{q}_i))$, is due to FAVOR+ and is quantified in [15]. On the other hand, the approximation $H$ follows from:

**Inter-channel distortion.** *The probability that inter-channel attention distorts the intended signal of the dot-product by a factor outside $[1 - \alpha, 1 + \alpha]$ has various upper bounds, most notably decaying exponentially w.r.t. $D\alpha^2 \cos^2(\angle(\overline{k}_j^{(u,n)}, \overline{q}_i^{(u,n)}))/(NM-1)^2$. See Appendix D for the full theorem.*

### 4.1 FAVOR+S: Computing self-attention in superposition

We first discuss separately how to use a one-dimensional grid to carry out either the multiplication ($\times$) or the construction of $A$ in superposition. Finally, the integration into a 2D grid will be shown.

**Placing multiple queries in superposition.** We set up channels $1, \ldots, M$ by simultaneously generating a superposition in the construction of $A_s$ and of the queries to be applied. To avoid inter-channel attention we superpose value-key tensor products, i.e., we do not construct tensor products between superposed values and superposed keys:

$$S_i = \underbrace{\left[ \sum_{j=1}^{L} \sum_{u=1}^{M} v_j^{(u)} \phi(k_j^{(u)})^T \right]}_{A_s} \times \phi(\sum_{t=1}^{M} q_i^{(t)}) = \sum_{j=1}^{L} \sum_{u=1}^{M} v_j^{(u)} \left( \phi(k_j^{(u)})^T \phi(\sum_{t=1}^{M} q_i^{(t)}) \right) \qquad (13)$$

$$\overset{P}{\approx} \sum_{j=1}^{L} \sum_{u=1}^{M} v_j^{(u)} \exp\left(\langle k_j^{(u)}, \sum_{t=1}^{M} q_i^{(t)} \rangle / \sqrt{D}\right) \overset{H}{\approx} \sum_{j=1}^{L} \sum_{u=1}^{M} v_j^{(u)} \exp\left(\langle k_j^{(u)}, q_i^{(u)} \rangle / \sqrt{D}\right) \qquad (14)$$

$$= \sum_{u=1}^{M} \underbrace{\sum_{j=1}^{L} v_j^{(u)} \exp\left(\langle k_j^{(u)}, q_i^{(u)} \rangle / \sqrt{D}\right)}_{\text{unnormalized } o_i \text{ of channel } u}. \qquad (15)$$

We obtain a superposition of bound output values; hence, the cost of computing $A \times \phi(q_i)$ for all $i$ is amortized across channels. However, the construction complexity of $A_s$ is increased $M$-fold to $\mathcal{O}(LDR \cdot M)$, hence the complexity per protected channel remains at $\mathcal{O}(LDR)$.

**Constructing value-key tensor products in superposition.** Next, we demonstrate a value-key tensor product ($A_s$) shared across all channels, but with a mere $\mathcal{O}(LD(R+N))$ setup complexity. In $\mathcal{O}(LND)$ we compute $\sum_{q=1}^{N} v_j^{(q)}$ and $\sum_{w=1}^{N} k_j^{(w)}$ for all $j$. These are then (re)used to build $A_s$.

$$S_i^{(n)} = \underbrace{\left[ \sum_{j=1}^{L} (\sum_{q=1}^{N} v_j^{(q)}) \phi(\sum_{w=1}^{N} k_j^{(w)})^T \right]}_{A_s} \times \phi(q_i^{(n)}) = \sum_{j=1}^{L} \sum_{q=1}^{N} v_j^{(q)} \left( \phi(\sum_{w=1}^{N} k_j^{(w)})^T \phi(q_i^{(n)}) \right) \qquad (16)$$

$$\overset{P}{\approx} \sum_{j=1}^{L} \sum_{q=1}^{N} v_j^{(q)} \exp\left(\langle\sum_{w=1}^{N} k_j^{(w)},\, q_i^{(n)}\rangle/\sqrt{D}\right) \overset{H}{\approx} \sum_{j=1}^{L} \sum_{q=1}^{N} v_j^{(q)} \exp\left(\langle k_j^{(n)},\, q_i^{(n)}\rangle/\sqrt{D}\right) \quad (17)$$

$$= \underbrace{\sum_{j=1}^{L} v_j^{(n)} \exp\left(\langle k_j^{(n)},\, q_i^{(n)}\rangle/\sqrt{D}\right)}_{\text{unnormalized } o_i \text{ of channel } n} + \underbrace{\sum_{q\neq n} \sum_{j=1}^{L} v_j^{(q)} \exp\left(\langle k_j^{(n)},\, q_i^{(n)}\rangle/\sqrt{D}\right)}_{\text{noise in separate protected subspace}}. \quad (18)$$

The output contains the $n^{th}$ channel together with noise. The operation $A_s \times \phi(q_i^{(n)})$, which takes $\mathcal{O}(LDR)$, must be repeated $N$ times to produce outputs for all $N$ channels, causing a bottleneck.

**Simultaneous superposition of queries and value-key tensor products using a 2D grid.** Finally, we combine the two previously described approaches to encode the superposition channels in a 2D grid of size $N \times M$. We multiply a constant matrix ($A_s$) with features derived from a superposition of queries to get the superposition vector $S_i^{(n)}$

$$S_i^{(n)} = \underbrace{\left[\sum_{j=1}^{L} \sum_{u=1}^{M} \left(\sum_{q=1}^{N} v_j^{(u,q)}\right) \phi(\sum_{w=1}^{N} k_j^{(u,w)})^T\right]}_{\text{construct } A_s \text{ in } \mathcal{O}(LMD(R+N))} \times \underbrace{\phi(\sum_{t=1}^{M} q_i^{(t,n)})}_{\text{construct } \forall i,n \text{ in } \mathcal{O}(LND(R+M))}. \quad (19)$$

Computing the multiplication $\times \ \forall i, n$ takes $\mathcal{O}(LNDR)$. If we set $M{=}N$, we can evaluate $S_i^{(n)} \ \forall i, n$ using only $\mathcal{O}(LNDR + LN^2D)$ instead of the usual $\mathcal{O}(LDR \cdot N^2)$. Thus, one may achieve a speedup of $\mathcal{O}(\min(\sqrt{N^2}, R))$ compared to FAVOR+. Since $R$ is normally in the hundreds [15], we can assume improvements of $\mathcal{O}(\sqrt{N^2})$ for reasonably large $N^2 = M \cdot N$. Eq. (19) simplifies to:

$$S_i^{(n)} = \sum_{j,u,q} v_j^{(u,q)} \left(\phi(\sum_w k_j^{(u,w)})^T \phi(\sum_t q_i^{(t,n)})\right) \overset{P}{\approx} \sum_{j,u,q} v_j^{(u,q)} \exp\left(\frac{\langle \sum_w k_j^{(u,w)},\, \sum_t q_i^{(t,n)}\rangle}{\sqrt{D}}\right) \quad (20)$$

$$\overset{H}{\approx} \sum_{j=1}^{L} \sum_{u=1}^{M} \sum_{q=1}^{N} v_j^{(u,q)} \exp\left(\langle k_j^{(u,n)},\, q_i^{(u,n)}\rangle/\sqrt{D}\right) \quad (21)$$

$$= \underbrace{\sum_{u=1}^{M} \sum_{j=1}^{L} v_j^{(u,n)} \exp\left(\frac{\langle k_j^{(u,n)},\, q_i^{(u,n)}\rangle}{\sqrt{D}}\right)}_{\text{unnormalized } o_i \text{ of channel } (u,n)} + \underbrace{\sum_{q\neq n} \sum_{u=1}^{M} \sum_{j=1}^{L} v_j^{(u,q)} \exp\left(\frac{\langle k_j^{(u,n)},\, q_i^{(u,n)}\rangle}{\sqrt{D}}\right)}_{\text{noise in separate protected subspace}}. \quad (22)$$

### 4.2 Integrating FAVOR+S into MIMOFormer

As is apparent in Eq. (19), the query superposition is along a different axis ($M$) than the key and value superpositions ($N$). The output of attention, however, is only superposed along a single axis ($M$). To be able to set up superpositions along both axes ($M$ and $N$) at the next layer, we require all channels (i.e., keys, queries, and values) in separation, i.e., not superposed, at the interface between MIMOFormer layers. We present two variants of MIMOFormer with different speedups.

The first computes in superposition exclusively during the attention mechanism. The individual tokens of the channel $(n, m)$ are directly retrieved from $S_i^{(n)}$ by unbinding with the key $\tilde{a}^{(n,m)} = a^{(n,m)}$, and the remaining computational steps within FAVOR+S are performed separately.

The second (and faster) MIMOFormer instance additionally performs the concatenation, the linear layer, as well as the MLP in superposition (shown in Figure 3). Unlike in the first variant, the skip connection around the attention block must account for the introduced superposition. To allow a potential embedding dimension mismatch, we instantiate two different sets of randomly drawn bipolar keys: one for the skip connection and post-MLP unbinding, and one for FAVOR+S binding. All keys are frozen during training; it is up to the trainable weights in the linear layer after concatenation to find the relationship between the binding and unbinding keys.

The function $\phi$ in the self-attention block consists of an $R{=}256$ dimensional projection and a ReLU activation [15]. Appendix C provides theoretical justification for using ReLU in $\phi$ and its benefits for MIMOFormer. Empirically, ReLU shows better numerical stability than unbiased softmax FAVOR+.

# 5 Empirical Results

This section evaluates the proposed MIMONets on various model architectures and benchmarks. Appendix E and Appendix F describe the experimental setup for MIMOConv and MIMONets, respectively. All experiments are repeated five times with a different random seed. We report the mean and standard deviations of accuracy to account for variability in training.

## 5.1 MIMOConv

**CIFAR10 and CIFAR100.** Our main baseline, which we adapt to MIMOConv, is a reproduced WideResNet-28-10 [11], i.e., a 28-layer CNN with the number of feature maps of each convolutional layer enlarged by $10\times$. Moreover, as a stronger baseline, we include the isometry regularization in the training of WideResNet-28-10, and call the resulting network WideIsoNet-28-10. Both baselines demand 5.251 GMACs (Giga multiply-accumulate operations) per sample. MIMOConv's inference complexity per sample is $5.335$ GMACs for $N$=1, $2.667$ GMACs for $N$=2, and $1.334$ GMACs for $N$=4; hence, we get a $\approx N\times$ speedup despite not accelerating the first and last layer. See Appendix E.

Table 1 shows the accuracy of the static MIMOConv, which is exclusively trained to support either 1, 2, or 4 channels. The MIMOConv with one channel ($N$=1) outperforms both baselines, which may be attributed to regularizing effects of the key-value binding. MIMOConv with $N$=2 channels still outperforms WideResNet-28-10, while reducing the inference complexity by $2\times$. The complexity can be further reduced by increasing the number of superpositions to $N$=4 at a slight accuracy drop of $\leq 3.18\%$, compared to WideResNet-28-10.

Next, we evaluate the dynamic partitioning of the superposition channels to select a speed-accuracy operating point instantaneously, which is a main feature of our approach and sets it apart from other static approaches that opt for a fixed performance point like model downsizing, quantization aware training, and pruning (see Appendix A for a discussion). We set up a model with four channels, but evaluate its performance in different configurations: a fast (4 inputs/pass), a normal (2 inputs/pass), and a slow mode (1 input/pass). The fast mode maps each input to one channel; the medium mode distributes two inputs over pairs of channels; and the slow mode uses all channels for the same input. The models are trained on 80% of the batches in fast mode and on 20% of the batches in slow mode. Appendix E provides more details on the trade-off between fast and slow mode training. As Table 1 shows, a single dynamic model can seamlessly switch between operation points while maintaining a high accuracy compared to the static models ($\leq 1.82\%$ drop).

Table 1: Average accuracy (%) of WideResNet-28-10 variants and our MIMOConv. Static models are trained to process $N$ inputs in one pass, speeding up inference by $N\times$. Dynamic models are trained with a variable number of inputs ($N$=1–4), and can process a variable number of inputs per pass. We report the average accuracy $\pm$ the standard deviation over five runs with different seeds.

| | CIFAR10 | | | CIFAR100 | | |
|---|---|---|---|---|---|---|
| # inputs/pass | 1 | 2 | 4 | 1 | 2 | 4 |
| WideResNet-28-10 | $96.82^{\pm0.06}$ | n.a. | n.a. | $81.62^{\pm0.07}$ | n.a. | n.a. |
| WideIsoNet-28-10 | $97.31^{\pm0.11}$ | n.a. | n.a. | $82.38^{\pm0.20}$ | n.a. | n.a. |
| MIMOConv static ($N$=1) | $97.49^{\pm0.08}$ | n.a. | n.a. | $83.19^{\pm0.17}$ | n.a. | n.a. |
| MIMOConv static ($N$=2) | n.a. | $96.93^{\pm0.13}$ | n.a. | n.a. | $82.30^{\pm0.19}$ | n.a. |
| MIMOConv static ($N$=4) | n.a. | n.a. | $95.58^{\pm0.23}$ | n.a. | n.a. | $78.44^{\pm0.30}$ |
| MIMOConv dynamic ($N$=1–4) | $97.13^{\pm0.11}$ | $96.41^{\pm0.14}$ | $95.43^{\pm0.07}$ | $82.52^{\pm0.09}$ | $80.48^{\pm0.08}$ | $78.19^{\pm0.10}$ |

The detailed ablation study in Appendix E makes the following findings: (1) isometry regularization improves accuracy for any number of channels; (2) training MIMOConv for more epochs closes the performance gap to the single-input baseline; (3) an appropriate number of feature maps (32 or 64) in the first layer stabilizes training; and (4) the binding keys can be frozen during training without loss (whereas unbinding keys are never frozen).

**MNIST and SVHN.** Figure 4 compares MIMO-Conv with DataMux [13] on the MNIST dataset. Even with a trivial downsizing for fair comparison from a 28-layer very-wide ($10\times$) ResNet to a 10-layer narrow ($1\times$) network, MIMOConv scales much better to high superposition channels ($N$) than DataMUX does. Indeed, our model shows an accuracy of $80.4\%$ against their $52.9\%$ in case of $N$=16 superposition channels (highest number of channels reported by DataMUX for vision tasks), despite being computationally cheaper ($0.47$ MMAC/s vs. $0.65$ MMAC/s). Also, DataMux's binding overhead results in a mere $1.35\times$ reduction in MACs compared to our $10.9\times$ as $N$ goes from 1 to 16. Ergo, our method scales better in accuracy and throughput as $N$ increases.

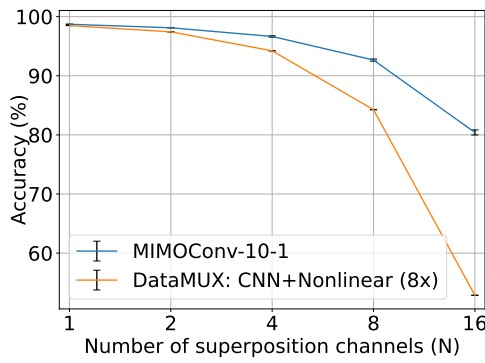

Figure 4: Classification accuracy (%) on MNIST for downsized model.

Finally, we tested MIMOConv on the SVHN dataset. Despite limited hyperparameter tuning, MIMOConv achieves a high accuracy of $97.17\%$ ($N$=1), and can maintain its performance with larger superpositions ($97.05\%$ and $96.84\%$ for $N$=2 and $N$=4, respectively).

## 5.2 MIMOFormer

**LRA.** We evaluate MIMOFormer on five tasks from LRA [14], and compare against the vanilla Transformer [24] and the Performer [15] using FAVOR+ attention with ReLU projection. Moreover, we also consider wide Transformer variants [26], consisting of only one layer but as many heads as their deep counterparts. Task-specific architectures and training hyperparameters are kept the same for the Performer and the MIMOFormer (see Appendix F).

Table 2: Test accuracy (%) on the long range arena (LRA). MIMOFormer uses an equal number of query superpositions ($M$) and value-key superpositions ($N$), i.e., $N$=$M$. Computation in superposition is performed either in attention only (att.) or in both attention and MLP (att.+MLP). $L$ is the number of layers, $H$ the number of heads, and $^*$ indicates curriculum learning.

| | ListOps | Text | Retrieval | Image | Pathfinder | Avg. |
|---|---|---|---|---|---|---|
| **Deep models** | $L$=6, $H$=8 | $L$=6, $H$=8 | $L$=4, $H$=4 | $L$=3, $H$=4 | $L$=4, $H$=8 | |
| Transformer [24] | 36.37 | 64.27 | 57.46 | 42.44 | 71.40 | 53.39 |
| Performer [15] | 18.01 | 65.40 | 53.82 | 42.77 | 77.05 | 51.41 |
| Performer (reproduced) | $38.94^{\pm0.23}$ | $65.70^{\pm0.31}$ | $81.58^{\pm0.18}$ | $40.14^{\pm0.86}$ | $73.82^{\pm0.78}$ | $60.04^{\pm0.47}$ |
| MIMOFormer ($N$=2, att.) | $38.08^{\pm0.21}$ | $65.00^{\pm0.28}$ | $79.37^{\pm0.81}$ | $38.21^{\pm0.63}$ | $72.36^{\pm0.54}$ | $58.61^{\pm0.49}$ |
| MIMOFormer ($N$=2, att.+MLP) | $37.65^{\pm0.33}$ | $64.39^{\pm0.22}$ | $76.02^{\pm0.27}$ | $33.85^{\pm0.55}$ | $67.98^{\pm0.47}$ | $55.98^{\pm0.37}$ |
| MIMOFormer ($N$=4, att.) | $37.22^{\pm0.33}$ | $64.59^{\pm0.14}$ | $60.99^{\pm9.06}$ | $28.16^{\pm0.08}$ | $55.50^{\pm4.95}$ | $49.29^{\pm2.91}$ |
| MIMOFormer ($N$=4, att.)$^*$ | $37.64^{\pm0.73}$ | $64.46^{\pm0.15}$ | $74.38^{\pm0.82}$ | $30.52^{\pm0.77}$ | $67.10^{\pm0.45}$ | $54.82^{\pm0.58}$ |
| MIMOFormer ($N$=4, att.+MLP) | $17.74^{\pm0.63}$ | $60.71^{\pm5.14}$ | $72.20^{\pm0.28}$ | $24.01^{\pm0.47}$ | $50.33^{\pm0.16}$ | $45.00^{\pm1.34}$ |
| **Wide models** | $L$=1, $H$=48 | $L$=1, $H$=48 | $L$=1, $H$=16 | $L$=1, $H$=12 | $L$=1, $H$=32 | |
| Performer (reproduced) | $39.40^{\pm0.51}$ | $65.73^{\pm0.32}$ | $83.67^{\pm0.25}$ | $41.67^{\pm0.44}$ | $74.11^{\pm0.33}$ | $60.93^{\pm0.37}$ |
| MIMOFormer ($N$=2, att.) | $38.90^{\pm0.53}$ | $65.39^{\pm0.18}$ | $81.27^{\pm0.28}$ | $40.25^{\pm0.21}$ | $73.51^{\pm0.23}$ | $59.86^{\pm0.29}$ |
| MIMOFormer ($N$=2, att.+MLP) | $37.59^{\pm0.17}$ | $64.64^{\pm0.25}$ | $78.30^{\pm0.32}$ | $36.69^{\pm0.76}$ | $68.22^{\pm0.18}$ | $57.09^{\pm0.34}$ |
| MIMOFormer ($N$=4, att.) | $37.71^{\pm0.24}$ | $64.22^{\pm0.14}$ | $74.99^{\pm0.36}$ | $35.43^{\pm0.60}$ | $69.52^{\pm0.40}$ | $56.37^{\pm0.35}$ |
| MIMOFormer ($N$=4, att.)$^*$ | $37.68^{\pm0.36}$ | $64.56^{\pm0.25}$ | $76.37^{\pm0.50}$ | $35.53^{\pm0.48}$ | $73.37^{\pm0.22}$ | $57.50^{\pm0.36}$ |
| MIMOFormer ($N$=4, att.+MLP) | $18.52^{\pm0.98}$ | $63.53^{\pm0.12}$ | $74.30^{\pm0.26}$ | $26.54^{\pm0.28}$ | $56.33^{\pm0.17}$ | $47.84^{\pm0.36}$ |

Owing to an improved training setup, our replicated deep and wide Performer baselines substantially outperform the results reported in [14] (see Table 2). Moreover, MIMOFormer enables accurate computation in superposition for both deep and wide attention models. The performance drop is less pronounced in wide models (only $1.07\%$ drop compared to Performer with $N$=2, att.), which may be attributed to the larger number of heads, increasing the effective dimension ($D_{\text{tot}} = H \cdot D_{\text{head}}$).

When computing both attention and the MLP in superposition (att.+MLP), we observe better scaling (in $N$) for wide models. Also, MIMOFormer reduces the gap to the baseline as the number of epochs increases (see Appendix F).

To stabilize training in the case of $N$=4, we implemented a curriculum training procedure where the number of superpositions is reduced to $N'$=$N/2$ during a warmup phase (1/6th of the training steps), improving the average accuracy of MIMOFormer in both wide and deep models.

Comparing against a reproduced DataMUX [13], MIMOFormer (att.) outperforms it on ListOps (38.08% vs. 30.54% accuracy) when using models of similar size and $N$=2, see Appendix F.

**Synthetic sequence modeling.** Table 3 reports the accuracy on two synthetic sequence modeling tasks, which Transformer alternatives such as S4 [27] have difficulties solving [12]. On these more nuanced NLP tasks, the accuracy of Data-MUX [13] drops to 20.04% and 6.06% for $N$=2 despite significant efforts in training, while MIMOFormer, at a score

Table 3: Accuracy (%) on synthetic sequence modelling.

| Architecture | Attention | Associative recall [12] | Induction head [12] |
|---|---|---|---|
| Transformer | Softmax | $98.48^{\pm1.87}$ | $100^{\pm0.0}$ |
| Performer | FAVOR+ | $96.32^{\pm6.26}$ | $31.58^{\pm33.67}$ |
| MIMOFormer (N=2, att.) | FAVOR+ | $96.52^{\pm2.79}$ | $99.40^{\pm0.13}$ |
| MIMOFormer (N=2, att.) | DPFP [25] | $93.64^{\pm12.66}$ | $98.56^{\pm0.86}$ |
| DataMUX (N=2) [13] | Softmax | $20.04^{\pm1.72}$ | $6.06^{\pm2.24}$ |

of 96.52% and 99.40% respectively, succeeds. We attribute this difference in performance to attention score blurring in DataMux, discussed in Appendix F. Contrastingly, our method converges to exact attention without blurring. It is versatile and can be adjusted to other linear Transformers such as DPFP [25], achieving a score of 93.64% and 98.56%.

## 6 Related Work

So far, superposition principles have been applied in order to store the weights of multiple models in a single neural network [28–30], to circumvent catastrophic forgetting in continual learning [31, 32], and to render symbolic reasoning tractable [33]. To address privacy concerns when running remote inference, single inputs were bound with random channels to implement pseudo-encryption [34]. Recently in [35], HRR was used to define an unconventional version of self-attention, whose attention scores are processed to a diagonal matrix. The value vectors are scaled according to their importance in the sequence instead of being combined in a weighted sum. In contrast to us, none of these works superpose multiple inputs into a data structure to speed up computation.

In [19, 20], an ensemble of CNN models was fit into one network. However, by only broadcasting a single input over the channels and by averaging all the outputs, this approach collapses to a single-input-single-output (SISO) network. On the contrary, we explore using protected channels for *different* inputs at inference, resulting in an actual multiple-input-multiple-output (MIMO) network.

There has also been a line of work to accelerate Transformers using inputs in superposition [13] [36]. DataMux [13] claims to retain high performance for language understanding tasks, even when using up to 40 inputs in superposition. However, none of the reported tasks require attention layers at all [37]. In Section 5.2 we show failure of their method when actual attention is required (see also Appendix F). MUX-PLMs [36] improves on DataMux with contextual binding and replaces token prefixes with unbinding keys, but does not address the blurry attention mechanism. In contrast to DataMUX and MUX-PLMs, our work approximates true attention and our theoretical derivations show convergence to actual dot-product attention as the dimension of attention projections increases, giving us an even stronger case for applicability to large language models.

## 7 Conclusion

We present MIMONets that simultaneously process multiple inputs by performing computation in superposition. MIMONets bind arbitrary inputs with high-dimensional keys, which projects them to orthogonal subspaces that, together with near-isometric subfunctions, guarantee low interference through all nonlinear layers. Unbinding with (learned) keys can safely retrieve information on individual channels. We provide two MIMONets instances, MIMOConv and MIMOFormer, that show the effectiveness of computation in superposition through two dominant operations in neural network architectures: convolution and attention. Further investigations could explore the MIMO-capability of architectures that contain additional nonlinearities (e.g., max-pooling) and use different input modalities. MIMONets could be suitable candidates to accelerate dynamically and on-demand the inference of foundation models [38].

## Acknowledgement

This work is supported by the Swiss National Science foundation (SNF), grant 200800. We thank Dario Bolli for conducting initial experiments and Aleksandar Terzić for helping with ablation studies.

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
