# OpenReview forum: "MIMONets: Multiple-Input-Multiple-Output Neural Networks Exploiting Computation in Superposition"
_NeurIPS.cc/2023/Conference — NeurIPS 2023 poster_

### Official Review · Reviewer_3r5d · 2023-07-06

**Soundness:** 3 good
**Presentation:** 2 fair
**Contribution:** 3 good
**Rating:** 5
**Confidence:** 2

**Summary:**

This study introduces Multiple-Input-Multiple-Output Neural Networks (MIMONets) that can process multiple inputs simultaneously, reducing computational cost. Two types of MIMONets, MIMOConv for CNNs and MIMOFormer for Transformers, are presented. MIMOConv can handle multiple image inputs with minimal accuracy loss, while MIMOFormer effectively calculates attention scores for two concurrent inputs. These models offer a dynamic balance between accuracy and processing speed, using a fixed set of parameters.

**Strengths:**

MIMONets significantly improve the processing speed by handling multiple inputs simultaneously, reducing the computational cost per input. hey can be applied to various neural network architectures, including CNNs and Transformers.

**Weaknesses:**

1. The concept of superposition can lead to interference between inputs, which may affect the model's accuracy.
2. The integration of variable binding mechanisms and transformations for holistic processing may increase the complexity of the model, potentially making it harder to implement and understand.
3. While the speed of processing is improved, there is a noted drop in accuracy when handling multiple inputs, which may not be suitable for applications requiring very high precision.

**Questions:**

1. Can the MIMONets approach be generalized to other types of neural network architectures beyond CNNs and Transformers?

**Limitations:**

Not applicable.

---

> ### Author Rebuttal · Authors · 2023-08-09
>
> We would like to thank you for your time and feedback. However, it is unclear how the sparse set of stated weaknesses led to a reject decision. In particular, weakness 2 applies to most innovations that are yet to be established. Weaknesses 1 and 3 coincide, are thoroughly addressed in the paper (for instance, Section 5.1 and Appendix E), and do not apply to the developed central use case of dynamic inference. We have responded to your remarks in more detail below and would appreciate an increased rating or further discussion.
>
> >“[Weakness 1] The concept of superposition can lead to interference between inputs, which may affect the model's accuracy.”
>
> Although this is true, it is rather a limitation of compute-in-superposition than a weakness of the paper. We bring up the issue of interference at multiple points and in fact come up with mitigation techniques such as isometry regularization and high hidden dimension. Specifically, the ablation study in Section 5.1 (with additional experiments and insights in Appendix E) explores such mitigation techniques to suppress the emergence of interference. Finally, to avoid a trade-off between accuracy and speedup we developed the concept of dynamic inference. In particular, we showed that one can enable a model to compute more quickly (via superposition) while still retaining the usual accuracy in the slow mode.
>
> >“[Weakness 3] While the speed of processing is improved, there is a noted drop in accuracy when handling multiple inputs, which may not be suitable for applications requiring very high precision.”
>
> Although a drop in accuracy is discernible, our method clearly improves on the state of the art Murahari, Vishvak, et al. “DataMUX: Data multiplexing for neural networks”, which was awarded second place in the 2022 Bell Lab Prize. Indeed, our comparison on MNIST, two synthetic language tasks, and a subtask of LRA (see Figure R1 and Table R1) shows the following: for high superpositions (16x) our CNN method outperforms theirs 80.4% to 52.9% on MNIST (the only vision dataset they report on) while being computationally cheaper, our Transformer (2x) outperforms theirs on ListOps 38.08% to 30.54%, and our method does not fail on synthetic language benchmarks which require faithful attention (scoring 96.52% and 99.40%) while DataMUX does (scoring 20.04% and 6.06%). Finally, as is quantified in Table 1 of the paper for dynamic inference, our method enables a single model with fixed parameters to be run at different accuracy-throughput operating points. In particular, at normal speed (N=1), the model performs as accurately as the baseline. This is a unique case where one obtains essentially free lunch, high accuracy and high throughput are guaranteed and can be balanced at will. Since there is no viable alternative to obtain such instantaneous switching between accuracy-throughput operating points within a fixed set of model parameters, and since we outperform state of the art, we do not see the observed drop in accuracy as a dealbreaker.
>
> >“[Weakness 2] The integration of variable binding mechanisms and transformations for holistic processing may increase the complexity of the model, potentially making it harder to implement and understand.”
>
> In terms of computational complexity, the integration of variable binding mechanisms via binding and unbinding operations is inconsequential, amounting to 0.008%--0.031% of the total MACs for MIMOConv and 0.06%--0.14% for MIMOFormer (see Table R2 and Table R3 in the pdf).
>
> Regarding understandability of variable binding, in fact we chose to go with some of the most well-defined binding/unbinding operations (see Table A1 in the paper) of vector-symbolic architectures [8--10]. Vector-symbolic-architectures propose frameworks for constructing symbolic data structures through key-value binding of distributed representations. The construction of this data structure is transparent thanks to the use of explicit binding/unbinding operations with well-defined properties. These properties allow us to compose neural representations on-the-fly as opposed to learning how to compose them from scratch. While it is less well explored to process the resulting data structure using **nonlinear** neural transformations, we believe the choice of well-established binding/unbinding operations leads to a more transparent architecture by design.
>
> >“[Question] Can the MIMONets approach be generalized to other types of neural network architectures beyond CNNs and Transformers?”
>
> CNNs and Transformers are the most used DNN architectures. Furthermore, in Table 2 of the paper we report both on Transformers with superposition applied to MLPs (att.+MLP) and without MLPs in superposition (att.). Also, CNNs are essentially constrained MLPs with weight-sharing and limited connectivity. Since unconstrained MLPs do not require the locality principles of CNNs, binding mechanisms in much higher-dimensional space can be employed making it easier by decreasing interference via the Blessing of Dimensionality. To further convince the reviewer of the wide applicability of our method, we demonstrate in Table R1 of the pdf that our proposed method of superposition for attention is not restricted to FAVOR+ from the Performer, but instead widely applicable to other linear transformers such as DPFP (Deterministic Parameter-Free Projection) from I. Schlag et al. “Linear transformers are secretly fast weight programmers". To summarise, we have shown our method to work for CNNs, MLPs, and Transformers. Although we do not provide empirical results on other structures such as RNNs and Graph Neural Networks, the theoretical arguments are general and explored in more details in the Appendix (e.g., A.2, A.3.).

---

> > ### Comment · Reviewer_3r5d · 2023-08-19
> >
> > As the author addressed most of the concerns, I will increase the score to 5.

---

> > > ### Author Response · Authors · 2023-08-21
> > >
> > > We are glad that we could address your concerns and thank you for increasing your score.

---

### Official Review · Reviewer_zbz3 · 2023-07-07

**Soundness:** 3 good
**Presentation:** 3 good
**Contribution:** 3 good
**Rating:** 6
**Confidence:** 2

**Summary:**

The main content of the article is about MIMONets, which are multiple-input-multiple-output neural networks that exploit computation superposition. By using fixed-width distributed representations in vector-symbolic architectures, MIMONets can represent a variable number of inputs in a data structure and process them holistically with nonlinear neural transformations. This leads to a significant reduction in computational burden per input and offers a dynamic trade-off between accuracy and throughput. The article presents two instances of MIMONets (MIMOConv and MIMOFormer), which apply the concept of computation in superposition to convolutional neural network (CNN) and Transformer architectures, respectively. Empirical evaluations show that MIMONets achieve significant speedups while maintaining high accuracy.

**Strengths:**

Strengths:
1. The method cleverly combines multiple inputs into a single sample for inference, reducing the computational cost. This is highly meaningful for practical applications.
2. The method achieves promising results in both CNN-based and Transformer-based structures, indicating its versatility and applicability.

**Weaknesses:**

Weaknesses:
1. The experimental datasets and network applications in the study are relatively limited. It would be beneficial to apply the method to more datasets, such as ImageNet, and explore its effectiveness on a wider range of networks.
2. When the number of stacked samples is 4, a noticeable performance drop is observed.

**Questions:**

1. In lines 86 and 87, why is it stated that n and $x^{(1)}$ are orthogonal?
2. I don't fully understand how Dynamic Inference is performed.
3. Are unbinding keys shared among all data or does each sample have a corresponding unbinding key?

**Limitations:**

I believe the limitations of the method mainly manifest in two aspects:
1. Performance significantly decreases when a large number of samples are stacked.
2. It remains uncertain whether similar performance can be maintained when applying the method to a wider range of structures.

---

> ### Author Rebuttal · Authors · 2023-08-09
>
> >“The experimental datasets and network applications in the study are relatively limited. It would be beneficial to apply the method to more datasets...”
>
> We value the reviewer's proposal to conduct additional experiments on other tasks. We addressed this by adding results on MNIST (see Figure R1 of the pdf) and two synthetic language tasks (“associative recall” and “induction head” reported on in Table R1). These tasks have been found to be challenging for language models without attention, such as S4 (Fu, et al. “Hungry Hungry Hippos: Towards Language Modeling with State Space Models”), but are easily solved by our method with accuracy 96.52% and 99.40% respectively. This is a strong indicator of the fidelity of our attention in superposition.
>
> Unfortunately, due to licensing issues, we were not able to use some datasets such as ImageNet. We also want to stress that this paper is partly of theoretical nature with many insights on the qualitative behavior of error-terms together with quantitative bounds (see Appendix). The mathematical bounds are especially powerful for large scale models and show vanishing interference in these cases. Given the successful results on smaller benchmarks we plan to extrapolate and experimentally validate on large language models in future.
>
> >“When the number of stacked samples is 4, a noticeable performance drop is observed.”
>
> Although a drop in accuracy is discernible, our method clearly improves on the state of the art DataMUX; see our comparison on MNIST, two synthetic language tasks, and a subtask of LRA (in Figure R1 and Table R1). Indeed, for high superpositions (N=16) our CNN outperforms theirs 80.4% to 52.9% on MNIST (their only vision dataset) while being computationally cheaper, our Transformer (N=2) outperforms theirs on ListOps 38.08% to 30.54%, and our method does not fail on the challenging synthetic language benchmarks, which require faithful attention (scoring 96.52% and 99.40%) while theirs does (scoring 20.04% and 6.06%). This is because DataMUX blurs attention scores by sharing them between superpositions. Finally, as is quantified in Table 1, for dynamic inference, our method enables **a single model** with fixed parameters to be run at different accuracy-throughput configurations: at normal speed (N=1) the model performs as accurate as the baseline.
>
> >“In lines 86 and 87, why is it stated that n and x(1) are orthogonal?”
>
> In the case of Hadamard binding $a^{(1)} \oslash a^{(1)} \odot x^{(1)} = x^{(1)}$ and hence the random noise vector $n = a^{(1)} \oslash a^{(2)} \odot x^{(2)}$ is a random vector unaffected by $x^{(1)}$. As a high-dimensional random vector it is almost orthogonal to a fixed vector $x^{(1)}$ with high probability. This “Blessing of Dimensionality” is explored in more depth for vectors of Rademachers in Appendix A.2.
>
> >“I don't fully understand how Dynamic Inference is performed.”
>
> Thanks for the feedback. With it at the center of our motivation we will add further explanation as follows:
>
> To illustrate the idea of dynamic inference, suppose only two superposition channels are used with binding keys $a^{(1)}, a^{(2)}$ and unbinding keys $\tilde{a}^{(1)}, \tilde{a}^{(2)}$. We already know how the model performs standard computation in superposition (see (1) - (4) in paper). Let us thus examine how a network with the same parameters can instead be used as an ensemble-method with higher accuracy, but lower throughput. A superposition is established of twice the same input $x^{(1)}$:
>
> $$s = a^{(1)} \odot x^{(1)}  +  a^{(2)}  \odot x^{(1)} $$
>
> After applying the deep neural network $f_\theta$ to the superposition, we may unbind as
>
> $$ \tilde{a}^{(1)} \oslash f_\theta(s) \approx \tilde{a}^{(1)} \oslash f_\theta\left(a^{(1)} \odot x^{(1)}\right) +  \tilde{a}^{(1)} \oslash f_\theta\left(a^{(2)} \odot x^{(1)}\right) \approx f_\theta\left(x^{(1)}\right) + \tilde{a}^{(1)} \oslash f_\theta\left(a^{(2)} \odot x^{(1)}\right). $$
>
> and
>
> $$\tilde{a}^{(2)} \oslash f_\theta(s) \approx \tilde{a}^{(2)} \oslash f_\theta\left(a^{(1)} \odot x^{(1)}\right) +  \tilde{a}^{(2)} \oslash f_\theta\left(a^{(2)} \odot x^{(1)}\right) \approx \tilde{a}^{(2)} \oslash f_\theta\left(a^{(1)} \odot x^{(1)}\right) + f_\theta\left(x^{(1)}\right). $$
>
> After averaging the two expressions, we get
>
> $$ \frac{1}{2}\left( \tilde{a}^{(1)} \oslash f_\theta(s)  + \tilde{a}^{(2)} \oslash f_\theta(s) \right) \approx f_\theta\left(x^{(1)}\right) + n$$
>
> where $n$ is a random noise vector and $f_\theta\left(x^{(1)}\right)$ is approximated as an average of two predictions. Owing to the introduction of stochasticity by the binding and unbinding process these predictions are decorrelated, i.e., each superposition channel is processed to some degree differently.
>
> >“Are unbinding keys shared among all data or does each sample have a corresponding unbinding key?”
>
> The unbinding keys are independent of data and depend only on the index of the superposition channel.
>
> >“[Limitations] Performance significantly decreases when a large number of samples are stacked.”
>
> As described above, the performance decrease of MIMOConv is significantly less pronounced than that of the SOTA. But it is true that there is a limit to the number of superposition channels that can be employed, mostly depending on the size of the hidden dimension.
>
> >“[Limitations] It remains uncertain whether similar performance can be maintained when applying the method to a wider range of structures.”
>
> CNNs and Transformers are the most used DNN architectures. MLPs are used in superposition within our Transformers. To further address the reviewers concerns, in Table R1 we demonstrate our method for attention to not be restricted to FAVOR+, but instead to be widely applicable to other linear transformers such as DPFP from Schlag et al. “Linear transformers are secretly fast weight programmers". Finally, the theoretical arguments are even more general and explored in more details in the Appendix (e.g., A.2, A.3.).

---

> > ### Comment · Reviewer_zbz3 · 2023-08-17
> > **reply to author's feedback**
> >
> > Thanks Reply. I recognize your work, but I still have some doubts and suggestions.
> > 1. Thank you very much for explaining dynamic inference in detail. I want to confirm again: Does Dynamic Inference refer to a single model that can infer 1 to N images?
> > 2. It is better to verify the results on a more convincing dataset. I can understand that this article may be more analytical and theoretical. But if there is a lack of more convincing verification, it is like a lack of soul.

---

> > > ### Author Response · Authors · 2023-08-21
> > >
> > > >”Thanks Reply. I recognize your work, but I still have some doubts and suggestions.”
> > >
> > > Dear reviewer. We are glad that you recognize our work. We address your remaining question and doubt/suggestion.
> > >
> > > >”Thank you very much for explaining dynamic inference in detail. I want to confirm again: Does Dynamic Inference refer to a single model that can infer 1 to N images?”
> > >
> > > Yes, that’s correct. Dynamic Inference refers to a single model that can infer 1 to N images (at once by putting them in superposition) at varying degrees of accuracy.
> > >
> > > >”It is better to verify the results on a more convincing dataset. I can understand that this article may be more analytical and theoretical. But if there is a lack of more convincing verification, it is like a lack of soul.”
> > >
> > > Despite the emphasis on theoretical analysis, our method has been shown to work on various meaningful datasets. State of the art CNNs, such as our baseline, are still challenged by CIFAR100. Furthermore, Long Range Arena is one of the most extensively reported dataset collections covering a wide range of tasks such as understanding long mathematical expressions, classifying and compressing (long) natural text documents, classifying natural images, and using visual-spatial reasoning to determine the path-connectedness of points.
> > >
> > > Following your request, we now provide additional results on the street view house number (SVHN) dataset. Despite the limited time (3 days) and hyperparameter tuning, MIMOConv achieves a high accuracy of 97.17% (N=1), and can maintain the performance with larger superpositions (97.05% and 96.84% for N=2 and N=4, respectively).

---

### Official Review · Reviewer_A964 · 2023-07-08

**Soundness:** 3 good
**Presentation:** 3 good
**Contribution:** 3 good
**Rating:** 6
**Confidence:** 4

**Summary:**

This paper proposes a novel method named multiple-input-multiple-output (MIMO) neural networks, which aims to achieve simultaneous inference for several inputs together by mixing them into one input. To this end, the authors invent one method to first encode the inputs, whose output can be decoded to give separate outputs. The proposed method is evaluated on both convolutional and attention models on CIFAR datasets and LRA framework.

**Strengths:**

The proposed method to mix several inputs is novel and interesting. Also, the authors conducted simple analysis and several experiments.

**Weaknesses:**

The method is only verified on some small datasets like CIFAR. Also, the results in Table 1 seem to give a significant accuracy drop for multiple input cases, compared to using a single input. Also, it would be better if the author could provide more results on other tasks, like object detection or segmentation.

**Questions:**

Could the author provide more explanation on how to apply the proposed method in practice?

**Limitations:**

The authors did not provide limitations. I think the practical application of the proposed method can be limited.

---

> ### Author Rebuttal · Authors · 2023-08-09
>
> >“The method is only verified on some small datasets like CIFAR […] it would be better if the author could provide more results on other tasks […]”
>
> We appreciate the reviewer's suggestion to conduct additional experiments on other tasks. We addressed this by adding results on MNIST (see Figure R1 of the pdf) and two synthetic language tasks (“associative recall” and “induction head” reported on in Table R1) in addition to the benchmarks present in the paper (CIFAR10/CIFAR100 and LRA). The mentioned synthetic language tasks have been found to be challenging for language models without attention, such as S4 (see Y. Fu, Dao, et al. “Hungry Hungry Hippos: Towards Language Modeling with State Space Models”), but are easily solved by our method (in superposition) with an accuracy of 96.52% and 99.40% respectively. This is a strong indicator of the fidelity of our attention in superposition.
>
> It is also important to note that this paper is partly of theoretical nature with many insights on the qualitative behavior of error-terms together with quantitative bounds, see the extensive Appendix. The mathematical bounds are especially powerful for large scale models and show vanishing interference in these cases. Given the successful results on smaller benchmarks we plan to extrapolate and experimentally validate on large language models in future.
>
> >“the results in Table 1 seem to give a significant accuracy drop for multiple input cases, compared to using a single input”
>
> Although a drop in accuracy is discernible, our method substantially improves on the state of the art Murahari, Vishvak, et al. “DataMUX: Data multiplexing for neural networks”; see our comparison on MNIST, two synthetic language tasks, and a subtask of LRA in Figure R1 and Table R1. In fact, in terms of accuracy, for high superpositions (N=16) our CNN method outperforms theirs 80.4% to 52.9% on MNIST (the only vision dataset they report on) while being computationally cheaper; our Transformer (N=2) outperforms theirs on ListOps 38.08% to 30.54%, and our method does not fail on the challenging synthetic language benchmarks which require faithful attention (scoring 96.52% and 99.40%) while theirs does (scoring 20.04% and 6.06%). This is because DataMUX blurs attention scores by sharing them between superposed sentences. Finally, as is quantified for dynamic inference in Table 1 of the paper submission, our method enables **a single model** with fixed parameters to be run at different accuracy-throughput operating points. In particular, at normal speed (N=1) the model performs as accurate as the baseline. This is a unique case where one obtains essentially free lunch, high accuracy and high throughput are guaranteed and can be balanced at will.
>
> >“Could the author provide more explanation on how to apply the proposed method in practice?”
>
> Excellent request, due to space limitations we did not explore the practical motivation to great extent. However, the following will be added to the next revision:
>
> Think of large language models which cost enormous amounts of money to train and run, require real-time response, and whose usage fluctuates heavily over time. By training such a model for dynamic inference a provider could ensure to service all its costumers no matter the demand, albeit at the cost of a (hopefully) unnoticeable drop in performance. Alternatively, an autonomous system on a tight memory and power-budget might need a higher accuracy in critical situations. Owing to memory-constraints, having multiple models (with different energy consumption) in memory might not be feasible and incurs additional data-transfer costs due to switching. MIMONet allows a framework where these otherwise incompatible demands can be fulfilled.
>
> >“The authors did not provide limitations. I think the practical application of the proposed method can be limited.”
>
> We agree and will add a more dedicated limitations section. It will read as follows:
>
> - MIMONets make use of the Blessing of Dimensionality, that with high probability exponentially many (in dimension D) vectors are almost orthogonal. Although the components of MIMONet are made near isometric through regularization, a certain number of (hidden) dimensions is still necessary. This naturally limits MIMONets to large (oftentimes over-parametrized) models or models employing low-rank decompositions.
>
> - The number of inputs that can be superposed without incurring heavy losses in accuracy is limited given a fixed neural network due to increasingly strong interference between the superposition channels.
>
> - The proposed superposition capable attention mechanism converges to faithful attention (without interference between channels) as the embedding dimension increases, but at the price of only a speedup of $N$ when using $N^2$ superposition channels. Being built on linearized attention such as FAVOR+, it further inherits all their benefits (linear scaling) and drawbacks (limited parallelization and increased memory accesses for autoregressive training (see Section 3.1 in Hua, Dai, Liu, et al. “Transformer Quality in Linear Time”). On the other hand, trivial superposition would yield a speedup of $N^2$ instead, but at the cost of blurring the attention scores with each token-token score summarizing attention in all superposition channels at once. Such models employing blurry attention are limited to application where imprecise “summarizing” information suffices.
>
> Regarding the last point we demonstrate in Table R1 of the pdf that our proposed method of superposition for attention is indeed not restricted to FAVOR+ from the Performer, but instead widely applicable to other linear transformers such as DPFP (Deterministic Parameter-Free Projection) from I. Schlag et al. “Linear transformers are secretly fast weight programmers".

---

> > ### Comment · Reviewer_A964 · 2023-08-21
> > **Thanks for the authors' response**
> >
> > Dear Authors,
> >
> > Thanks for the response and discussion. I would prefer to keep my score based on it.
> >
> > Best.

---

### Official Review · Reviewer_b2eJ · 2023-07-12

**Soundness:** 3 good
**Presentation:** 3 good
**Contribution:** 3 good
**Rating:** 7
**Confidence:** 4

**Summary:**

UPDATE: scores updated based on rebuttal.

This paper proposes a method (MIMONets) for multiplexing multiple independent samples in superposition in such a way that one can train neural networks to simultaneously process those samples in training and inference. The method is adapted both for CNNs and Transformers and there is empirical evaluation for both that show the viability of the method.

**Strengths:**

The paper is well written and easily understandable, and the method is relatively well grounded in foundations of previous works and new analysis.

**Weaknesses:**

The authors claim that this is the first time this has been done, but they fail to mention some related work such as Murahari, Vishvak, et al. "DataMUX: Data multiplexing for neural networks." Advances in Neural Information Processing Systems 35 (2022): 17515-17527. Please review that work and discuss similarities and differences, and if possible conduct experimental comparison. The DataMUX paper claims an impressive 40 samples multiplexing for the transformers, while MIMONet uses just 4. At least, the CNN part of the MIMONet work seems to give better empirical results, though.

One of the strongest motivators for the MIMONets is the ability to “dynamically” scale computation vs. accuracy at inference run-time using the same trained weights by changing the way input and output is processed, and by creating in-network ensembles if more resources are available. The authors should center the motivation more in this area unless they can have a strong argument to the “static” case.

Regarding the static case, there is less convincing results. For example, since there is a drop in accuracy with multiplexing 2-4 samples compared to 1, the comparison should be done with a smaller baseline with the same accuracy, and the authors should include full FLOPS figures for the models, including the bind/unbind operation.

Also, for the static case, some comparison methods include pruning, quantization aware training, compression. It would be needed to compare these in terms of intOPS/FLOPS and power consumption estimates using models of the same accuracy.

MIMOConv

For the MIMOConv model, the authors should include the mathematical description of bind/unbind including all dimensions and indexes.

Table 1 does not clearly state the FLOPS / sample for each of the rows.

Table 1 does not discuss what would be the FLOPS size of a baseline model that reaches the same accuracy as e.g., N=4 MIMOConv models.

MIMOFormer

The authors should add some more information on the Transformer model, for example, in the Figure 3 it seems every layer does bind/unbind. This seems different from the CNN models, so that authors should describe this more in detail and give more background to these choices. What is the complexity addition in FLOPS from these bind/unbind operations?

Table 2 (MIMOFormer results) is not fully clear what is the computational complexity (per sample) for each of the rows. For example, do the MIMOFormer models have higher complexity in FLOPS because of the bind/unbind operations? I assume that MIMOFormer N=4 is almost 4x more efficient pers sample than N=1, but it would be great to report full model FLOPS/sample for each of the rows.

Table 2 seems to hint that the +MLP version performs worse (especially for N=4). But do I understand correctly that it also has less FLOPS since the -MLP version repeats the same MLP for each of the superimposed samples? This should be described. Would the +MLP version perform better if MLP was bigger?

Table 2 is missing comparison to a non-MIMO Performer baseline that would have the same FLOPS complexity per sample.

In Table 2 there seems to be some tasks, such as retrieval, where N=4 seems to make the performance drop dramatically. Could the authors discuss these?


**Questions:**

See above

**Limitations:**

Comparison for DataMUX missing, some apples-to-oranges comparisons in the experimental section.

---

> ### Author Rebuttal · Authors · 2023-08-09
>
> >“[the authors] fail to mention some related work […] Please review that work and discuss”
>
> Thank you for pointing out the DataMUX paper. Given its importance, we discuss key differences and compare against it empirically on additional benchmarks (see pdf) in the global response. In short, DataMUX does not design a dedicated binding method for CNNs, and their Transformers not only introduce additional coarse-grained “sentence summary” tokens but also compute fuzzy attention by sharing attention scores over sentences. Empirically, this is visible in their failure on the synthetic language benchmarks “associative recall” and “induction head”, which require faithful attention (scoring 20.04% and 6.06%) while our method succeeds (scoring 96.52% and 99.40%). Moreover, our Transformer with N=2 outperforms DataMux with N=2 on ListOps, 38.08% vs. 30.54%. Also, on high superpositions (N=16) our CNN method outperforms theirs (80.4% vs. 52.9% on MNIST) while being computationally cheaper. We will include these findings in a revision.
>
> >“[the strongest motivator for the MIMONets is the ability to “dynamically” scale computation vs. accuracy at inference run-time […] the authors should center the motivation more in this area”
>
> Thanks for the feedback. We tried to highlight dynamic inference as our strongest advantage and most significant innovation, but we will improve on the wording and put further emphasis on it.
>
> >“Regarding the static case, there is less convincing results”
>
> We agree that dynamic inference is more convincing than static, as we add the capability of high-speed inference without sacrificing accuracy at normal speed. The rationale for including static results is to display the very slim drop in performance of each operating point against the static models. We will make that clearer. Even so, we outperform state of the art alternatives as mentioned above.
>
> >“[the static case should be compared to pruning, quantization aware training, and compression] in terms of intOPS/FLOPS…”
>
> We do not claim to outperform other throughput-increasing approaches like model downsizing, quantization, and pruning. However, in our opinion, we do not have to due to their lack of ability for dynamic accuracy-throughput selection. Furthermore, our approach is orthogonal to pruning, quantization, compression, and low-rank matrix decompositions. Each of them could be applied on top of our MIMO approach. However, for reasons of transparency, we include more detailed MACs in Table R2, R3, and Figure R1.
>
> >“For the MIMOConv model, the authors should include the mathematical description of bind/unbind including all dimensions and indexes.”
>
> Thank you for the suggestion; we will add the following:
>
> PWHRR is given by (with image tensors $x^{(k)} \in \mathbb{R}^{D \times W \times H}$ and binding key $a^{(k)} \in \mathbb{R}^D$)
>
> $$(a^{(k)} \odot x^{(k)}) {\scriptsize {:,w,h}} = a^{(k)} * x^{(k)}_{:,w,h}$$
>
> where $*$ is the usual circular convolution and $k$ indexes the superposition channel.
>
> Unbinding is performed by a linear layer (with hidden tensor $h \in \mathbb{R}^{D \times W \times H}$ and unbinding key $\tilde{a}^{(k)} \in \mathbb{R}^{d \times d}$)
>
> $$ (\tilde{a}^{(k)} \oslash h) {\scriptsize :,w,h} = \tilde{a}^{(k)} \cdot h_{:,w,h}$$
>
> where $\cdot$ is the usual matrix multiplication. Unbinding is applied after the global pooling to reduce MACs.
>
> >“In the Figure 3 it seems every layer does bind/unbind. This seems different from the CNN models, so that authors should describe this more in detail and give more background to these choices”
>
> Yes, this is indeed different to MIMOConv. In our proposed attention mechanism, we lay out the superposition channels along a 2D grid. We construct two separate superpositions along different axes, which is the only configuration s.t. attention scores are not shared between channels, i.e., remain accurate. Since the output of our attention layer is a single superposition along one of the two axes, we need to dismantle it for the next layer. Section 4.1 and 4.2 discussed these choices in greater detail.
>
> >“What is the complexity addition in FLOPS from these bind/unbind operations?”
>
> The relative complexity overhead in MACs is very low (0.06% to 0.14% depending on the configuration, see Table R3 in pdf).
>
> >“Table 2 is not fully clear what is the computational complexity (per sample) for each of the rows.”
>
> Thank you for the feedback. We added these results in Table R3.
>
> >“I assume that MIMOFormer N=4 is almost 4x more efficient per sample than N=1, but it would be great to report full model FLOPS/sample for each of the rows”
>
> This is the case. When examining Table R3 and neglecting the cost of K/Q/V-projections, N=4 att.+MLP is exactly 3.98 times faster than Performer. Not superposing before the K/Q/V-projection is a clear oversight. As is apparent from equation (18) in the paper, one could instead superpose before projecting without incurring losses. Future work could address this.
>
> >“[does +MLP have less FLOPs than -MLP, given it has lower accuracy?]”
>
> Yes, the -MLP version repeats the same MLP for each superposed sample. The additional data in Table R3 should clarify.
>
> >“Would the +MLP version perform better if MLP was bigger?”
>
> Nice point. The performance drop between +MLP and -MLP is significantly bigger in tasks with a low embedding/hidden dimension (e.g., Retrieval, Image, Pathfinder). This is also supported by our theoretical analysis, which indicates vanishing interference as the network increases in size. It also motivates extrapolation and experimental validation on large language models in the future.
>
> >“In Table 2 there seems to be some tasks, such as retrieval, where N=4 seems to make the performance drop dramatically. Could the authors discuss these?”
>
> Those are issues with training stability, as is apparent in the large standard deviation reported in this configuration. We could opt to discard outliers or report the median instead.

---

> > ### Comment · Reviewer_b2eJ · 2023-08-15
> > **Comment**
> >
> > Thank you for your rebuttal! One further comment. You say "Furthermore, our approach is orthogonal to pruning, quantization, compression, and low-rank matrix decompositions. Each of them could be applied on top of our MIMO approach."
> >
> > I think this is too strong statement without proof. Can you explain why you believe the gains from MIMO and e.g., quantization and compression are fully additive? Intuitively it feels like quantization, compression would make it harder to do the MIMO approach with good performance. Or put in another way, there might be high dependency between these approaches and combining them might not give fully additive gains.

---

> > > ### Author Response · Authors · 2023-08-17
> > >
> > > >”Thank you for your rebuttal! One further comment. You say "Furthermore, our approach is orthogonal to pruning, quantization, compression, and low-rank matrix decompositions. Each of them could be applied on top of our MIMO approach."
> > >
> > > >I think this is too strong statement without proof. Can you explain why you believe the gains from MIMO and e.g., quantization and compression are fully additive? Intuitively it feels like quantization, compression would make it harder to do the MIMO approach with good performance. Or put in another way, there might be high dependency between these approaches and combining them might not give fully additive gains.”
> > >
> > > Thank you for your comment. While we cannot state that the gains of MIMO and other methods are completely additive without experimental validation, we have qualitative insights as to why these methods do not compete for the same resources and consequently would synergize to some extent:
> > >
> > > - The Blessing of Dimensionality (see Appendix A.2) gives, in terms of dimensionality, an exponentially decreasing probability of interference for superpositions, even for (2-bit quantized) Rademachers. The extent to which these superpositions can be kept intact as linear layers act on them depends on the conditioning of the matrix (ideally nearly-isometric) not on the fidelity of its entries. As such we suspect that MIMOConv can be combined with quantization, pruning, etc.
> > >
> > > -  Regarding MIMOFormer, we can give quantitative insights. As is evident from Theorem 3 in Appendix D, the error bounds have no dependence on the precision of projection weights, but depend only on the embedding dimensionality, the size of keys and queries, and the angles between them. Consequently, quantization, pruning, etc. are not in competition with our approach and can be easily combined.
> > >
> > > Naturally, when combining different methods not only the gains but also the errors add up. However, with diminishing returns of each method we believe the combination of several to be most effective, especially given that our method is not competing with alternatives for the same resources of a model.
> > >
> > > If there are further questions or comments we are happy to discuss them.

---

> > ### Comment · Reviewer_b2eJ · 2023-08-15
> > **Comment**
> >
> > Thank you. You write "Those are issues with training stability, as is apparent in the large standard deviation reported in this configuration. We could opt to discard outliers or report the median instead.". It would be much better to understand the source of unstability and fix that instead. Any thoughts?

---

> > > ### Author Response · Authors · 2023-08-17
> > >
> > > >“Thank you. You write "Those are issues with training stability, as is apparent in the large standard deviation reported in this configuration. We could opt to discard outliers or report the median instead.". It would be much better to understand the source of unstability and fix that instead. Any thoughts?”
> > >
> > > Actually, we have to clarify. It is rather unreliable training than an issue of stability. The loss reaches a plateau with some random seeds, after which the model yields no meaningful predictions (i.e., random chance). According to your remarks we ran additional experiments and found a work-around.
> > >
> > > To improve training in the high superposition regime, we implemented a curriculum training procedure where the number of superpositions is reduced to N’=N/2 during a warmup phase (1/6th of the training steps). Afterwards, the number of superpositions is increased to the original value (N). This curriculum procedure improved the average accuracy of MIMOFormer (N=4, att.) from 60.99% to 74.38%, and reduced the standard deviation from 9.06% to 0.74%. We will add curriculum training results on complete LRA in the final version of the paper.
> > >
> > > Thank you for raising this point. We believe that curriculum learning is a valuable addition to our MIMONets.

---

> > > > ### Comment · Reviewer_b2eJ · 2023-08-17
> > > > **Thank you**
> > > >
> > > > Thank you for the additional experiments and the results. I will reflect all these in my score.

---

> > > > > ### Author Response · Authors · 2023-08-21
> > > > >
> > > > > We are happy to hear that our replies and revisions were able to address your concerns, and we thank you for increasing your score.

---

### Author Rebuttal · Authors · 2023-08-09

We thank the reviewers for their thoughtful feedback. We are encouraged that (b2eJ) found our method well-written and well-grounded in previous works. We are pleased that (b2eJ, A964) appreciated our new theoretical analysis and experimental evaluation. We are glad that (zbz3, 3r5d) agree with the versatility of our approach and that (b2eJ, zbz3) assess our results as promising, particularly for increasing throughput as noted by (zbz3, 3r5d). We appreciate that (b2eJ, zbz3) share our assessment that new applications are enabled by our work. In the following, we provide answers to the reviewer’s comments, which will be reflected in a revised version of the paper.

Regarding applications, we would like to remind the reviewers of the central innovation termed “*dynamic inference*”, where one can select on-the-fly an operating point of a given accuracy & throughput. As such, our method is only inadequately comparable with “*static*” approaches that opt for a fixed performance point like model downsizing, quantization aware training, and pruning. Furthermore, these other methods can be added on top.

To address questions by (b2eJ, A964, zbz3) we will add several clarifying paragraphs to the paper. Also, as (b2eJ) requested, we now include two Tables (R2, R3) which indicate the complexity of sublayers in MIMOConv and MIMOFormer respectively. Fitting the suggestion of (A964, zbz3) we conducted experiments on two additional datasets: MNIST and a synthetic language task. Finally, at the remark of (A964) we now include a dedicated limitations list.

We particularly thank (b2eJ) for pointing out the NeurIPS 2022 paper Murahari, Vishvak, et al. “DataMUX: Data multiplexing for neural networks”, awarded second place in the 2022 Bell Lab Prize. Shortly after submission we also discovered this work in the context of a patentability search. While there are some similarities between their methods and ours (e.g., both use Hadamard binding and unbind before the final readout layer), there are fundamental differences that distinguish them qualitatively and quantitatively. These are discussed in the following two points.

1\) As (b2eJ) puts it: “the CNN part of the MIMONet work seems to give better empirical results”. To strengthen this point, we conducted a direct comparison on the MNIST benchmark for which DataMUX was optimized, and report on the findings in Figure R1 of the pdf. Even with a trivial downsizing for fair comparison from a 28-layer very-wide (10x) MIMOConv to a 10-layer narrow (1x) MIMOConv, we scale much better to high superposition channels (N) than DataMUX does. Indeed, our model shows an accuracy of 80.4% against their 52.9% in case of N=16 superposition channels (highest #channels reported by DataMUX for vision tasks), despite being computationally cheaper (0.47 MMAC/s vs. 0.65 MMAC/s).

We attribute the improved performance to a set of innovations which we reiterate here: MIMOConv applies *position-wise binding*, thus retaining the locality property present in natural images and vital for CNNs, whilst as discussed by Murahari, Vishvak, et al. their primary binding does not. As a workaround they proposed binding via two layers CNNs each outputting 8 feature maps. The resulting (pixel-wise) superposition in a low-dimensional space (8-D) leads to high interference. Additionally to using an expensive binding mechanism, it also makes the first layer of the model 8 times as expensive no matter the number of superpositions. We are able to circumvent this issue by applying the first layer of the CNN *before* the pixel-wise binding, increasing the dimensionality of each pixel in an easy-to-understand manner. Another contribution is the use of *isometric neural networks* to further reduce interference during the processing of superposed images.

2\) Regarding Transformers, new experiments on LRA show that we outperform the Transformer variant of DataMUX on ListOps (38.08% vs. 30.54% accuracy) using models of similar size. Contrary to us, DataMUX blurs attention scores by sharing them between superposed sentences, and the introduction of additional global “sentence summary” tokens (w/o superposition) limits their approach to instances where imprecise “summarizing” information suffices. Notably, none of the tasks (token-level and sentence-level classification) they chose to report on requires attention layers at all; this is also discussed in M. Hassid et al. “How Much Does Attention Actually Attend”. As our experiments confirm (see Table R1), on more nuanced tasks in NLP like “associative recall” and “induction head”, their method drops to 20.04% and 6.06% for N=2, while ours, at a score of  96.52% and 99.40% respectively, succeeds. Despite investing significant efforts in the training of DataMUX, it cannot perform on these synthetic tasks. This is in line with the findings of Y. Fu, Dao, et al. “Hungry Hungry Hippos: Towards Language Modeling with State Space Models” which identify the lack of attention as the reason that the Structured State Space Sequence (S4) model is able to completely outperform state of the art in LRA, but is not feasible for large language models. In contrast to DataMUX, our work approximates true attention and our theoretical derivations *show convergence to actual dot-product attention* as the hidden dimension increases, giving us an even stronger case for applicability to large language models (for instance, GPT-3 uses embedding dimension 12,888, far exceeding the maximum of 512 we report on).

Finally, note that Linearized Transformers were able to decrease the complexity of dot-product attention from $O(L^2)$ to $O(L)$. Our MIMO-style is compatible with them and thus reduces the complexity from $O(L)$ further to $O(L/N)$. It achieves this while retaining the property that, in the limit of large embedding size, it converges *precisely* to quadratic attention.

We again thank the reviewers for their comments and look forward to entering into a more detailed discussion.

---

### Decision · Program_Chairs · 2023-09-21

**Decision:**

Accept (poster)

**Comment:**

The paper introduces the concept of Multiple-Input-Multiple-Output Neural Networks (MIMONets) that simultaneously process multiple inputs, resulting in a reduced computational burden. Two variants, MIMOConv (for CNNs) and MIMOFormer (for Transformers), are detailed, demonstrating the utility of the approach in both architectures. Empirical evaluations indicate that MIMONets can achieve notable speedups while preserving acceptable accuracies.

### Strengths:
* The idea of processing multiple inputs concurrently in order to significantly reduce computational cost during batching is interesting.
* The method's applicability to both CNN-based and Transformer-based structures demonstrates its versatility.
* Empirical evaluations provide insights into the trade-offs between computational efficiency and accuracy.

### Weaknesses:
* The paper lacks adequate comparison to related works like DataMUX and lacks proper reference to other foundational works based on MIMO (multiple input multiple output). See also [1] which speeds up ensembles via MIMO.
* Limited experimental datasets have been used such as only up to CIFAR100, narrowing the scope of the validation.
* Notable drop in accuracy is observed when handling multiple samples due to interference in superposition. This particularly happens with the stacking of 4 samples.
* Some issues in presentation clarity, specifically concerning the tables detailing the FLOPS/sample, and more information is desired about bind/unbind operations and their computational complexity.

Given the innovation and potential of MIMONets, juxtaposed against the cited weaknesses, especially around comparisons and detailed evaluations, I recommend accept. However, the authors should consider incorporating the feedback, particularly the comparison with relevant prior work, to strengthen their paper.

[1]: Training independent subnetworks for robust prediction. ICLR 2021. https://arxiv.org/abs/2010.06610